# BmCBP Catalyzes the Acetylation of BmApoLp-II Protein and Regulates Its Stability in Silkworm, *Bombyx mori*


**DOI:** 10.3390/insects14040309

**Published:** 2023-03-23

**Authors:** Yanmei Chen, Jiao Lv, Guowei Zu, Fan Yang, Jiasheng Geng, Zhengying You, Caiying Jiang, Qing Sheng, Zuoming Nie

**Affiliations:** Zhejiang Provincial Key Laboratory of Silkworm Bioreactor and Biomedicine, College of Life Sciences and Medicine, Zhejiang Sci-Tech University, Hangzhou 310018, China

**Keywords:** *Bombyx mori*, BmCBP, BmApoLp-II, acetylation, protein stability

## Abstract

**Simple Summary:**

*Bombyx mori* is an important economic insect and a model organism of Lepidoptera. Its hemolymph contains rich nutrient storage proteins, which participate in material transportation, immune regulation, and other physiological functions. Additionally, a large number of proteins in hemolymph can be acetylated. Apolipophorin-II is a kind of nutrition storage protein that can bind lipid substances. In the article, we confirm that the *Bombyx mori* acetyltransferase cyclic adenosine monophosphate response element binding protein (CREB) binding protein (CBP) can interact with the apolipophorin-II protein. Therefore, CBP protein, as an acetyltransferase, can catalyze the acetylation of apolipophorin-II protein and thus affect the stability of apolipophorin-II protein. Acetyltransferase CBP can affect the growth and development of insects. In general, the research on nutrition storage protein and CBP protein can develop a new direction for biological control and pest resistance in the future.

**Abstract:**

Acetylation is an important and reversible post-translational modification (PTM) of protein, which is involved in many cellular physiological processes. In previous studies, lots of nutrient storage proteins were found to be highly acetylated in silkworms, and acetylation can improve the stability of these proteins. However, the related acetyltransferase was not involved. In the present work, a *Bombyx mori* nutrient storage protein, apolipophorin II (BmApoLp-II), was further confirmed to be acetylated, and the acetylation could improve its protein expression. Furthermore, RNAi and Co-IP showed that the acetyltransferase BmCBP was found to catalyze the acetylation modification of BmApoLp-II, and thus affect its protein expression. Meanwhile, it was proved that acetylation could improve the stability of the BmApoLp-II protein by completing its ubiquitination. These results lay a foundation for further study on the mechanism of regulating nutrition storage and hydrolysis utilization of storage proteins by BmCBP and the acetylation in the silkworm *Bombyx mori*.

## 1. Introduction

Protein modification refers to the chemical modification of proteins after biosynthesis, also known as post-translational modifications (PTMs). PTMs of proteins mainly include methylation [1,2], lysine succinylation [3,4,5], O-GlcN acylation [6,7], phosphorylation [8,9,10], acetylation [4,8,11] and ubiquitination [12,13,14]. Acetylation, as a major PTM, is highly reversible and mainly occurs in lysine residue. It refers to the transfer of acetyl groups from acetyl CoA to the protein lysine ε-Amino side chain. It widely exists in various organisms and occupies a very important position in the life activities of organisms [15,16]. Many studies have shown that acetylation modification of proteins is involved in many life activity processes, including gene transcription, chromosome structure adjustment, cell signal transduction, protein stability, protein interaction, subcellular localization, cell apoptosis and so on [17,18,19]. Acetylation modification of proteins was first found in eukaryotic histones [20]. Acetylation of histones is affected by the dynamic balance of histone acetylases (HATs) and histone deacetylases (HDACs) [21]. With the in-depth study of histone acetylation modification and the progress of proteomics technology, it has been found that non-histones can also be acetylated and affected by HATs and HDACs [22,23].

Cyclic adenosine monophosphate response element binding protein (CREB) binding protein (CBP) and EP300 protein (P300) have extensive sequence homology and functional similarity. They are commonly referred to as CBP/P300 proteins, which define a unique family of HATs [24]. Acetyltransferase CBP/P300 has a strong catalytic function for acetylation in the cellular regulatory network. In addition, CBP/P300 can be used as a scaffold for recruiting other proteins, including transcription factors and co-activators [25], and several other reported HATs, such as nuclear receptor coactivator 1 (NCOA1), nuclear receptor coactivator 3 (NCOA3) and activating transcription factor 2 (ATF2) [26]. Although the function of CBP acetyltransferase has been confirmed, its research in insects is still very limited. It has been reported that 20-Hydroxyecdysone can promote CBP transcription and histone H3 lysine 27 (H3K27) acetylation in the model species of Lepidoptera, *Bombyx mori* [27,28]. The acetyltransferase CBP of the model organism *Drosophila* also has acetylation activity of H3K56 [29]. The importance of acetyltransferases CBP and P300 in living organisms may be determined by several factors: firstly, CBP/P300, as a transcriptional cofactor, plays a key role in transcriptional regulation [30]; secondly, CBP/P300 has strong acetyltransferase activity when it is overexpressed in cells, making it easier to identify its targets; and thirdly, the acetylation sites regulated by CBP and/or P300 have higher average stoichiometry [25].

Apolipoprotein is a kind of nutrition storage protein that can bind lipid substances [31]. The apolipophorin protein is a homologous protein of apolipoprotein. It is a multifunctional lipid protein in insects. There are three Apolipophorin proteins, including Apolipophorin-I, Apolipophorin-II and Apolipophorin-III (ApoLp-I, ApoLp-II and ApoLp-III) [32,33,34]. ApoLp-II/I is a homologous protein of insect apoB, belonging to the large lipid transfer protein superfamily [35]. It has the structural basis of a lipoprotein that acts as a lipid shuttle [36]. Different from ApoB, ApoLp-II/I will be decomposed into two proteins in insects: ApoLp-I and ApoLp-II. Besides lipid transport function, *Bombyx mori* ApoLp-II (BmApoLp-II) also has certain immune activity [37,38]. In previous studies, a large number of acetylated nutrient storage proteins were found in silkworms [39]. Acetylation can improve the stability of these proteins [33]. In this work, it has also been verified that acetylation can regulate the stability of the nutrient storage protein BmApoLp-II in silkworms. Furthermore, acetyltransferase BmCBP was found to be responsible for catalyzing the acetylation of the BmApoLp-II protein, thus regulating its protein expression. This work provides a new perspective for the studies on the acetylation function of non-histones in insects.

## 2. Materials and Methods

### 2.1. Cell Culture, cDNA and Plasmids

*Bombyx mori* ovarian cell line (BmN) was preserved in our laboratory and cultured in a 27 °C sterile incubator. The BmN cells were subcultured with SF-900 II medium (Gibco, California, USA, 10902088) containing 10% fetal bovine serum (Gibco, California, USA, 10091155) every 3–4 days. TRIzol (Invitrogen, California, USA, 15596018) was used to lyse the BmN cells and then extract the total RNAs. cDNA was synthesized using a reverse transcription kit (Takara Bio, Kyoto, Japan, RR037Q) according to the manufacturer’s instructions, and then stored at −80 °C. The pET-28a and pIEx-1-si-GFP plasmids were preserved in our laboratory.

### 2.2. The Construction of Eukaryotic/Prokaryotic Expression Vector 

The open reading frame (ORF) of the BmApoLp-II gene (GenBank Accession Number: AB640623.1) was amplified through the PCR method. The primers were listed as follows (the underlines were the *Bam*H I *and Xho* I restriction sites, respectively): 

Fp:5′-CCGGATCCGATAAATGCTCGTTGGCGTG-3′;

Rp:5′-CGCTCGAGTCTACGFCCTCTCGTTAGTCCATC-3′. 

The PCR reaction was carried out according to the following protocol: 94 °C for 2 min, followed by 40 cycles at 94 °C for 30 s, 66 °C for 30 s, and 68 °C for 60 s. Then the PCR product, which was purified by Agarose gel electrophoresis and a gel recovery kit (Axygen, California, USA, AP-GX-250), was digested by *Bam*H I/*Xho* I (Takara Bio, Kyoto, 1010A/1094A) and ligated with pIEx-1-si-GFP/pET-28a digested with the same restriction enzymes. The ligated products were transformed into *Escherichia coli* TG1 to construct the recombinant eukaryotic/prokaryotic expression vector pIEx-1-si-GFP-BmApoLp-II/pET-28a-BmApoLp-II. The recombinant plasmids were identified using PCR and double enzyme digestion, and then sent to Sangon Biotech (Shanghai, China) for further identification by sequencing.

### 2.3. Antibody Preparation

The constructed recombinant plasmid pET-28a-BmApoLp-II was transformed into *Escherichia coli* BL21 (DE3), and then the prokaryotic expression was induced by adding isopropyl β-D-1-thiogalactopyranoside (IPTG) to a final concentration of 1.0 mM. The inclusion body proteins were dissolved with 8 M urea and the recombinant BmApoLp-II protein was purified using nickel column affinity chromatography with a series of concentrations of imidazole (100 mM, 150 mM and 200 mM, respectively) according to the protocol described in this report [40]. After that, the eluent was added into a 50 kDa ultrafiltration tube for ultrafiltration, and finally, the purified protein was identified by SDS-PAGE analysis and used to prepare the antibody by HUABIO company (Hangzhou, China). ELISA and Western blotting were performed to identify the titer and specificity of prepared anti-BmApoLp-II antibody, respectively. 

### 2.4. Drug Treatment

In order to determine the effect of acetylation on the expression of BmApoLp-II protein in BmN cells, the cells were treated with different chemicals, including deacetylase inhibitor (TSA (MCE, New Jersey, USA, HY-15144), LBH589 (MCE, New Jersey, USA, HY-10224) or Cocktail (MCE, New Jersey, USA, HY-K0030)), acetylase inhibitor (C646 (MCE, New Jersey, USA, HY-13823) or A-485 (MCE, New Jersey, USA, HY-107455)), protein synthase inhibitor CHX (MCE, New Jersey, USA, HY-12320) and proteasome inhibitor MG132 (MCE, New Jersey, USA, HY-13259), respectively. The working concentrations of LBH589 or TSA used to treat cells were 0, 100, 200, 500, 1000 and 2000 nM, respectively. Similarly, a series of concentrations of C646 (0, 1, 2, 5, 10 and 20 μM, respectively) and A-485 (0, 10, 20 and 40 μM, respectively) were used to treat the cells, while the dosing volume of the cocktail used to treat cells was 0, 1, 2, 5, 10 and 20 μL, respectively. At 48 h post-drug treatment, the cells were lysed at 4 °C for half an hour and centrifuged at 12,000 rpm for 3 min. Then the supernatant was subjected to subsequent detection. To determine the effect of the acetylation modification on the accumulation or degradation of the BmApoLp-II protein, the cells were treated with protein synthesis inhibitor CHX and proteasome inhibitor MG132 at the working concentration of 10 mg/mL; meanwhile, 10 μL deacetylase inhibitor cocktail was added. In addition, the cells were treated with drugs for 0, 3, 6 and 9 h, respectively, to determine the time dependence.

### 2.5. Co-IP/IP Assays

The interaction between BmApoLp-II and BmCBP was verified by Co-IP (co-immunoprecipitation). Firstly, the BmN cells were transfected with pIEX-1-si-GFP-BmApoLp-II using FuGENE HD Transfection Reagent (Promega, Madison, USA, E2311), and 72 h post-transfection, the cells were lysed using a lysis buffer (Beyotime, Shanghai, China, P0013) at 4 °C for half an hour and then subjected to the Co-IP assay using the Dynabeads Protein G Immunoprecipitation Kit (Invitrogen, California, USA, 10007D) according to the protocol. Briefly, 4 μL anti-His antibody (Proteintech, Chicago, USA, 66005-1-Ig) or 50 μL rabbit-derived anti-BmCBP antibody prepared previously were diluted with 200 μL Ab binding washing buffer and incubated with immunomagnetic beads overnight, and then incubated with the cell lysate collected by centrifugation. The beads/proteins complexes were washed using a washing buffer three times, followed by a Western blotting detection using anti-His antibody, anti-BmCBP antibody, anti-GFP antibody (Proteintech, Chicago, USA, HRP-66031) and anti-Tubulin antibody (Proteintech, Chicago, USA, 66002-1-Ig), respectively. An IgG antibody (Proteintech, Chicago, USA, B900620) was used as a negative control. 

Furthermore, in order to verify the relationship between acetylation and ubiquitin in BmApoLp-II, pIEX-1-Si-GFP-BmApoLp-II and pIEX-1-Si-GFP-Ub were co-transfected into BmN cells. The target protein of BmApoLp-II was immunoprecipitated by IP kit and verified by Western blotting using a pan-acetylation antibody (PTM Biolabs, China, PTM-102) and anti-ubiquitin antibody (HUABIO, Hangzhou, China, ET1609-21), respectively.

### 2.6. RNAi Assay

A fragment of *Bombyx mori CBP* gene (*BmCBP*, GenBank Accession Number: XM_038015130.1) was amplified by PCR method. The PCR primers were listed as follows: Fp:5′-ATGGCCGATGAGCCGCCTAACAAGC-3′; Rp:5′-CGCCGCTGACCACACCTATG-3′. A Target Clone^TM^-Plus kit (TOYOBO, Osaka, Japan, TAK-201) was used to TA clone according to the protocol, and the recombinant T-vector pTA2-BmCBP, which contained the targeted fragment of *BmCBP*, was constructed. A linear DNA was designed by PCR using pTA-BmCBP as a template. Then the linear DNA and T7 RiboMAX™ Express RNAi System (Promega, Madison, USA, P1700) was used to synthesize the dsRNA specific to a particular target region of the *BmCBP*. The primers used were as follows: 

Primers without T7 promoter:

dsBmCBP-Fp:5′-AGGCGATACAAAACGGG-3′;

dsBmCBP-Rp: 5′-CCTGCGGTTGCGACT-3′. 

Primers with T7 promoter: 

dsBmCBP-T7-Fp: 5′-TAATACGACTCACTATAGGAGGCGATACAAAACGGG-3′;

dsBmCBP-T7-Rp: 5′-TAATACGACTCACTATAGGCCTGCGGTTGCGACT-3′. 

At the same time, a siRNA specific to the same target region of *BmCBP* was designed and synthesized by GenePharma (Shanghai, China). The dsRNA or siRNA was transfected into BmN cells for the knock-down of the *BmCBP* gene, and the acetylation level of histone H3K27 was detected to evaluate the knock-down efficiency. Next, the dsRNA/siRNA and pIEx-1-si-GFP-BmApoLp-II were co-transfected into BmN cells. At 72 h post-transfection, the cells were harvested and used to detect the expression and acetylation level of BmApoLp-II protein by Western blotting using anti-His antibody and pan-acetylation antibody, respectively.

### 2.7. qPCR

The quantitative PCR (qPCR) was used to detect the mRNA level of BmApoLp-II and *BmCBP*. The primers of qPCR for BmApoLp-II are as follows: Fp: 5′-GTACCGGCGATAGCATCGAA-3′; Rp: 5′-AGCAGCTTTTAGGGCATCGT-3′. The primers of qPCR for *BmCBP* are as follows: Fp: 5′-CCGTATTTTGAGGGCGACTTCTG-3′; Rp: 5′-TCTGCCTGCTTTCTTTTCTCCTCTT-3′. The primers of qPCR for *β-actin* are as follows: Fp: 5′-CGGCTACTCGACTTCTG-3′; Rp: 5′- CCGTCGGGAAGTTCGTAAG -3′. The qPCR was performed by using the GoTaq^®^qPCR Master Mix kit (Promega, Madison, USA, A6001). The *β-actin* gene was used as the internal control. Each group of samples were set up with three technical replicates and five biological repeats. The qPCR was performed by using the ABI7500 instrument. After the reaction, the relative mRNA level of the target gene was calculated by the 2^−ΔΔct^ method [41,42].

### 2.8. Western Blotting

The cells were lysed by cell lysis buffer for Western and IP (Beyotime, Shanghai, China, P0013) at 4 °C for half an hour, and the protein sample from the supernatant was obtained after centrifuging the cell lysate. The protein sample was prepared with 5× loading buffer and centrifuged after a 100 °C water bath. Then the protein sample was separated with 12.5% PAGE Gel. After electrophoresis, the gel was transferred to a polyvinylidene difluoride (PVDF) membrane (Roche, Basel, Switzerland, 65310600) on ice for 90 min at a 100 V voltage by Power Pac (Bio-Rad, Hercules, USA). After that, the PVDF membrane was sealed with 5% skim milk powder compounded by TBST for 2 h at room temperature. Next, the PVDF membrane was incubated with the corresponding primary antibody and secondary antibody at 4 °C for 4 h and 1 h, respectively. Finally, the membrane was treated with ECL Chromogenic solution (Beyotime, Shanghai, China, P0018M), and detected by using a Tanon 5500 chemiluminescence instrument. 

### 2.9. Statistical Analysis

The Image J (Version 1.8.0) software was used to obtain the grey scale of the protein band by Western blotting. The GraphPad Prism (Version 9.0) software was used for analyzing the statistically significant differences and creating the diagram. The differences between the groups were analyzed by one-way analysis and *t* test of variance. * *p* < 0.05, ** *p* < 0.01 and *** *p* < 0.001 indicate a difference, significant difference and extremely significant difference, respectively.

## 3. Results

### 3.1. Identification of Acetylation in BmApoLp-II Protein

First, the recombinant plasmid, pIEx-1-Si-GFP-BmApoLp-II, was constructed successfully (Figure 1A). It can make GFP protein and BmApoLp-II protein co-express in BmN cells without interference. Under the fluorescence microscope, it was found that GFP protein started to express at 12 h after transfection. With the increase of time, the protein expression also increased. At 48 h after transfection, the cell fluorescence was strong and the cell state was good (Figure 1B), suggesting a high overexpression of GFP and BmApoLp-II at this time. Therefore, the cells transfected for 48 h were collected and lysed to obtain the cell total proteins. The expression of BmApoLp-III was detected by Western blotting, and a specific band was found at the expected position (Figure 1C). At the same time, the expressed BmApoLp-II protein was isolated by IP with 6×His monoclonal antibody and detected by anti-His antibody/pan-acetylation antibody. The result indicated that the BmApoLp-II protein has high acetylation modification (Figure 1D). It is consistent with the previous acetylation identification by mass spectrometry [39]. 

### 3.2. Preparation of Polyclonal Antibody against BmApoLp- II Protein

There was an obvious specific band between 70–100 kDa by SDS-PAGE (Figure 2A), indicating that the BmApoLp-II protein was successfully expressed. After that, the recombinant protein was purified by nickel column affinity chromatography and ultrafiltration, and the BmApoLp-II protein with high purity was obtained (Figure 2B). New Zealand white rabbits were immunized with the purified protein to obtain rabbit polyclonal antibody against BmApoLp-II protein. The titer of the antibody reached to 1:16,000 (Figure 2C) detected by ELISA, and the specificity of the antibody was verified by Western blotting (Figure 2D). 

### 3.3. Acetylation Up-Regulates the Expression of BmApoLp-II Protein

First, we treated cells with different concentrations of deacetylase inhibitors trichostatin A (TSA) and panobinostat (LBH589). It was found that there was no significant change in the expression of BmApoLp-II compared with the co-expressed GFP protein (Figure 3A). However, when another acetylase inhibitor cocktail was used, the expression of the BmApoLp-II protein increased obviously along with the dosage of the cocktail (Figure 3B). Then, the cells were treated with acetylase inhibitors C646 and A485, respectively, and the expression of the BmApoLp-II protein indicated a significant downward trend after treatment with the two drugs (Figure 3C,D). Meanwhile, in order to detect the effect of acetylation modification on the expression of endogenous BmApoLp-II protein in BmN cells, cocktail, C646 and A485 were used to directly treat BmN cells, respectively. The expression of endogenous BmApoLp-II protein in the cells was identified by Western blotting using the prepared anti-BmApoLp-II polyclonal antibody. The results were consistent with the results above. The expression of endogenous BmApoLp-II protein also increased obviously along with the dosage of cocktail (Figure 3E) and decreased along with the dosage of the acetylase inhibitors C646 (Figure 3F) and A485 (Figure 3G). Therefore, these results indicate that acetylation can up-regulate the expression of BmApoLp-II protein. BmN cells treated with different concentrations of Cocktail, C646 and A485, respectively, were collected for qPCR analysis. The results showed that there was no statistically significant difference in the mRNA level of the BmApoLp-II gene under the effect of different concentrations of cocktail and C646 (Figure 3H,I); however, the mRNA level was up-regulated after A485 treatment (Figure 3J). It is speculated that A485 could up-regulate the expression of anti-apoptotic genes at the mRNA level [43], and BmApoLp proteins also have anti-apoptotic activity [23,44], requiring further studies. The cells treated with A485 showed a downward trend at the protein level for BmApoLp-II, however, there was an upward trend at the mRNA level. This indicates that changes in the acetylation level greatly influenced the expression of the BmApoLp-II protein at the post-transcriptional level.

### 3.4. BmCBP Interacts with BmApoLp-II Protein and Catalyzes Its Acetylation

The mRNA level of the *BmCBP* gene in BmN cells transfected with dsRNA or siRNA was detected by qPCR. It was found that the mRNA level of the *BmCBP* gene in BmN cells significantly decreased after dsRNA or siRNA transfection, and the interference effect of dsRNA was found to be better than that of siRNA (Figure 4A). It indicates that the dsRNA or siRNA can knock down the expression of the *BmCBP* gene. Furthermore, Western blotting analysis showed that the acetylation level of H3K27 in the cells was down-regulated at 48 h after transfection (Figure 4B). A previous report has confirmed that BmCBP can catalyze the acetylation of histone H3K27 [25], suggesting that dsRNA or siRNA can significantly affect the acetyltransferase activity of BmCBP by knocking down its gene expression. To further explore the relationship between the acetylation level of BmApoLp-II and the acetyltransferase BmCBP, dsRNA /siRNA and pIEx-1-Si-GFP-BmApoLp-II plasmid were co-transfected into BmN cells. The results showed that knock-down of the *BmCBP* gene by dsRNA or siRNA could down-regulate the acetylation level of the BmApoLp-II protein significantly (Figure 4C). Furthermore, it was verified that BmCBP could interact with the BmApoLp-II protein. The His-BmApoLp-II protein could be found in the sample immunoprecipitated with anti-BmCBP polyclonal antibody, while it could not be found in the sample immunoprecipitated with a negative control IgG (Figure 4D left). Meanwhile, the BmCBP protein could also be detected in the sample immunoprecipitated with anti-His monoclonal antibody (Figure 4D right). These results confirm that acetyltransferase BmCBP can catalyze the acetylation modification of BmApoLp-II protein in silkworms.

Furthermore, it was found that when the BmN cells were co-transfected with dsRNA or siRNA, the expression of the BmApoLp-II protein in the cells was significantly down-regulated (Figure 4F). The results above show that the transfected dsRNA or siRNA could decrease the acetylation level of the BmApoLp-II protein by inhibiting the acetyltransferase activity of BmCBP, thus reducing the expression of the BmApoLp-II protein in BmN cells. Therefore, based on the previous results, we suggest that BmCBP can regulate the expression of BmApoLp-II at the post-transcriptional level by changing its acetylation level.

### 3.5. Acetylation Could Improve the Stability of Protein in Cells

First, BmN cells transfected with pIEx-1-Si-GFP-BmApoLp-II were treated with CHX. The result showed that the expression of the BmApoLp-II protein showed a downward trend. However, the expression of the BmApoLp-II protein showed an upward trend in the cells treated with deacetylase inhibitor cocktail, which can maintain the acetylation state and stability of proteins and CHX simultaneously (Figure 5A). This indicates that acetylation prevents protein degradation to a certain extent. When the cells were treated with MG132, the BmApoLp-II protein was accumulated gradually and showed an upward trend expression, while in the cells treated with cocktail and MG123 simultaneously, it showed a more obvious upward trend expression (Figure 5B). This indicates that acetylation further increased BmApoLp-II protein accumulation in cells. The data above suggest that acetylation can improve the stability of protein in BmN cells.

### 3.6. Acetylation Could Compete for Ubiquitination to Stabilize BmApoLp-II Protein

Western blotting analysis showed that after the cocktail treatment to increase the acetylation level of BmApoLp-II, its ubiquitination level decreased, while after A485 treatment to reduce the acetylation level of BmApoLp-II, the ubiquitination level increased (Figure 6). This indicates that the acetylation of the BmApoLp-II protein can compete for its ubiquitination modification. From these results, it could be deduced that the acetylation modification may inhibit the ubiquitin-mediated proteasome degradation pathway by competing for its ubiquitination modification, thereby improving the stability of the BmApoLp-II protein.

## 4. Discussion

Acetylation modification, as a broad post-translational modification of proteins, is catalyzed by lysine acetyltransferase or lysine deacetylase to add or remove acetyl groups on histone and non-histone [45]. CBP/P300 has intrinsic histone acetyltransferase (HAT) activity due to its HAT domain and induces the acetylation of histone H3 and H4 [46]. At the same time, it is also widely involved in the acetylation regulation of non-histones and plays an important physiological role in organisms [47].

There are many nutrient storage proteins in the silkworm, such as apolipoprotein, storage protein and 30K protein. Our previous work found that these nutrient storage proteins contain multiple acetylation sites through nano-HPLC/MS/MS technology [39]. Meanwhile, previous studies have shown that acetylation modification could improve the protein stability of nutrient-storage proteins, including SP1 [48], SP2 [49], BmApoLp-III [33] and Bm30K-3 [23], which indicates that lysine acetylation may represent a common regulatory mechanism for the utilization of nutrients in silkworms [39]. However, the acetyltransferase which catalyzes the acetylation of these nutrient storage proteins in silkworms remains unknown.

It was demonstrated previously that CBP has the intrinsic activity of histone or non-histone acetyltransferase [50]. C646 and A485 are specific small molecule inhibitors of CBP [26]. In the present work, as the ortholog of CBP, the acetyltransferase BmCBP was found to catalyze the acetylation of nutrient storage protein BmApoLp-II and affect its protein expression. Firstly, BmApoLp-II protein was successfully expressed and proved to be acetylated by IP and Western blotting. Then it was found that deacetylase inhibitor cocktail upregulated the expression of overexpressed or endogenous BmApoLp-II. However, the other two inhibitors, TSA and LBH589, had no effect on the expression of BmApoLp-II. In our previous studies, TSA and LBH589 have been shown to increase the expression of SP2 [49], 30k-3 [23] and BmApoLp-III [33] in silkworms at the post-translational level, suggesting that deacetylation of different nutrient-storage proteins could be catalyzed by specific deacetylase; therefore, the deacetylation of BmApoLp-II is catalyzed by a specific deacetylase, needing further studies. Meanwhile, the acetylase inhibitors C646 and A485 could downregulate the expression of the BmApoLp-II protein. These drug treatments had no effect on the mRNA level of the BmApoLp-II gene through qRT-PCR, indicating acetylation could affect the expression of BmApoLp-II, and the effect occurs at the post-transcriptional level. Furthermore, RNAi and Co-IP showed that BmCBP could catalyze the acetylation modification of BmApoLp-II and regulate its protein expression. CHX and MG132 treatments further confirmed acetylation could improve the stability of BmApoLp-II. Finally, it was found that the increase of the acetylation level of BmApoLp-II could decrease its ubiquitination level, suggesting a competitive relationship between acetylation and ubiquitination of BmApoLp-II. Based on the results, it was speculated that acetyltransferase BmCBP could coordinate the acetylation and ubiquitination modification, and influence the protein stability of BmApoLp-II, thus regulating its expression at a post-translational level. 

Acetyltransferase P300/CBP can catalyze non-histone acetylation modification, exercising important biological functions. For example, activity inhibition and gene knock-down showed that P300/CBP can catalyze the acetylation modification of thousands of non-histone proteins, regulating Notch, Wnt, LKB-AMPK and TGF-β signaling pathways [25]. P300/CBP can regulate the function of non-histones by catalyzing acetylation and competing for ubiquitination, thereby improving protein stability. For example, in mouse lung epithelial cells, CBP-catalyzed acetylation can down-regulate the ubiquitination level of the FBXL19 protein, blocking the ubiquitin-proteasome pathway, thereby stabilizing the FBXL19 protein and improving its anti-inflammatory ability [51]. Similarly, CBP can promote tumor cell metastasis by catalyzing the acetylation of the DOT1L protein in colon cancer cells to improve their protein stability [52]. In the present work, we also found BmCBP could catalyze the acetylation of the BmApoLp-II protein and regulate its protein stability. As a nutrient-storage protein, BmApoLp-II was found to participate in lipid trafficking and also has an immunocompetent function [53]. Therefore, the acetylation of BmApoLp-II catalyzed by BmCBP might be involved in lipid utilization in silkworms. This study provides a further theoretical basis for exploring the storage and utilization of nutrition regulated by acetylation modification in the silkworm, *Bombyx mori*. 

## Figures and Tables

**Figure 1 insects-14-00309-f001:**
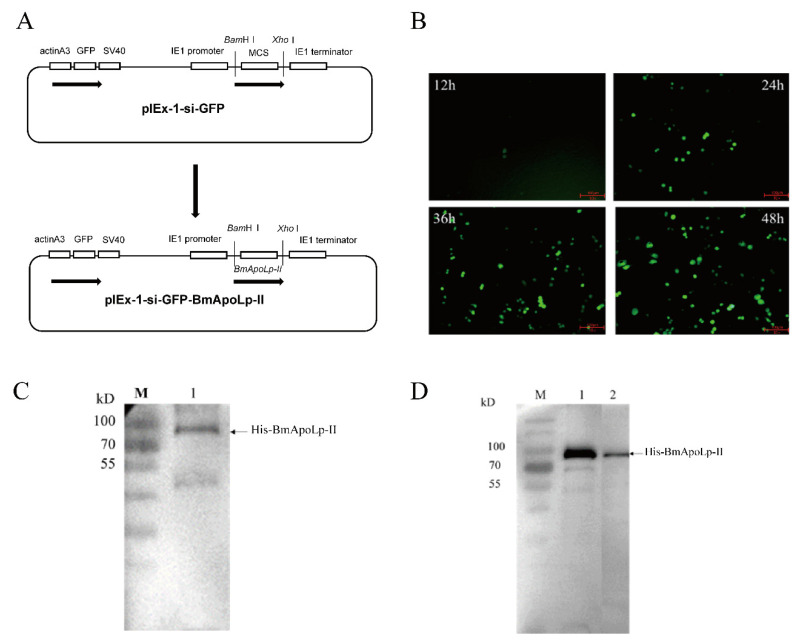
**Identification of acetylation in BmApoLp-II protein.** (**A**) The flowchart of pIEX-1-si-GFP-BmApoLp-II plasmid construction. (**B**) Expression of GFP co-expressed with BmApoLp-II in BmN cells. It was observed under a fluorescence microscope at different times after transfection. (**C**) Expression of BmApoLp-II fusion protein in BmN cells. It was identified by Western blotting using anti-His monoclonal antibody. (**D**) The immunoprecipitated BmApoLp-II protein was detected by Western blotting using a pan-lysine acetylation antibody. M: Pre-stained protein marker; 1: Detection by anti-His antibody; 2: Detection by pan-lysine acetylation antibody.

**Figure 2 insects-14-00309-f002:**
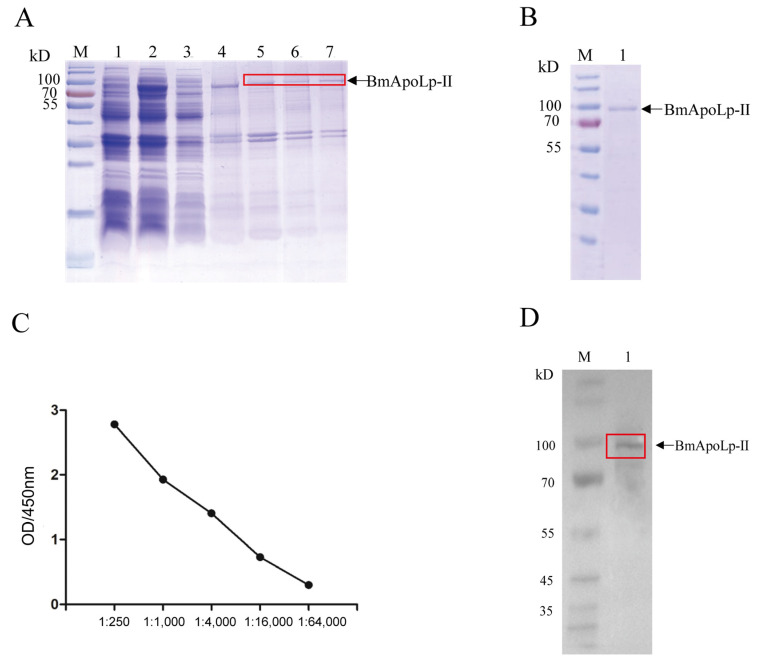
**Preparation of polyclonal antibody against BmApoLp-II protein.** (**A**) Prokaryotic expression and purification of BmApoLp-II. M: Pre-stained protein marker; 1: uninduced BL21; 2: induced BL21; 3: Supernatant of ultrasonic fragmentation; 4: Precipitation of ultrasonic fragmentation; 5: Elution by 100 mM imidazole; 6: Elution by 150 mM imidazole; 7: Elution by 200 mM imidazole. (**B**) Purification of BmApoLp-II by ultrafiltration. M: Pre-stained protein marker; 1: Purified BmApoLp-II protein. (**C**) Determination of the titer of anti-BmApoLp-II polyclonal antibody from rabbit serum. (**D**) Identification of the polyclonal antibody against BmApoLp-II. M: Pre-stained protein marker; 1: The lysis of BmN cells. Red box represents the band of BmApoLp-II protein.

**Figure 3 insects-14-00309-f003:**
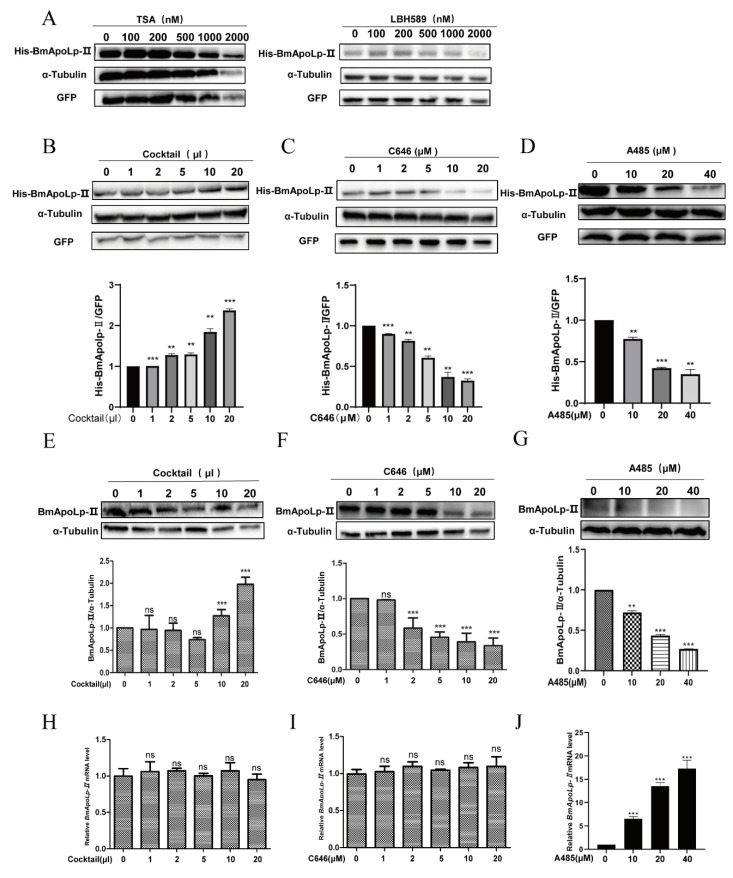
**Acetylation up-regulates the expression of BmApoLp-II protein.** (**A**) Compared with the co-expressed GFP protein, there was no significant effect on the expression of BmApoLp-II in BmN cells treated with different concentrations of TSA (left) or LBH589 (right). (**B**) Compared with the co-expressed GFP protein, the expression of BmApoLp-II protein increased significantly with the increase of cocktail usage. (**C**) Compared with the co-expressed GFP protein, the expression of BmApoLp-II showed a significant downward trend after C646 treatment. (**D**) Compared with the co-expressed GFP protein, the expression of BmApoLp-II showed a significant downward trend after A485 treatment. (**E**) The cells were treated with different concentrations of cocktail, and the expression of endogenous BmApoLp-II protein also showed an upward trend. (**F**) C646-treated cells showed a downward trend of endogenous BmApoLp-II protein expression. (**G**) A485-treated cells showed a downward trend of endogenous BmApoLp-II protein expression. (**H**) The transcriptional level of BmApoLp-II did not change meaningfully after treatment with cocktail. ns: no significance. (**I**) The transcriptional level of BmApoLp-II did not change meaningfully after treatment with C646. ns: no significance. (**J**) The treatment of A485 up-regulated the mRNA level of BmApoLp-II gene. The data represent the results of at least three independent experiments. ns: no significance, ** *p* < 0.01, *** *p* < 0.001.

**Figure 4 insects-14-00309-f004:**
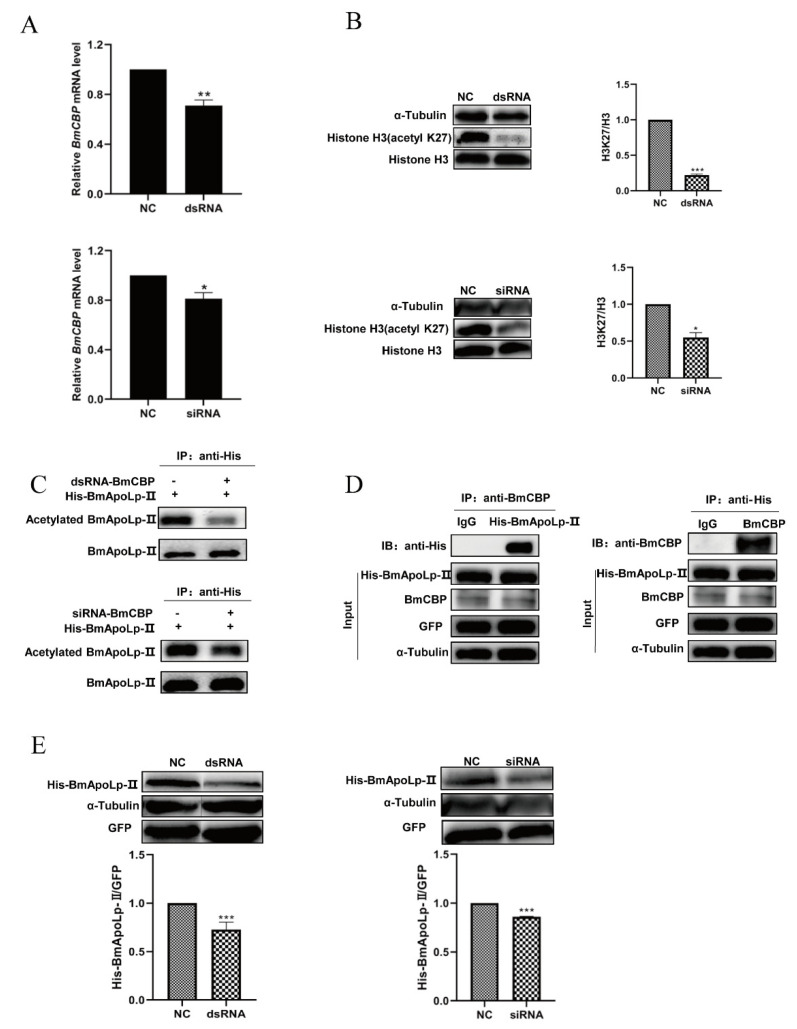
**BmCBP interacts with BmApoLp-II protein and catalyzes its acetylation.** (**A**) After dsRNA or siRNA transfection, the mRNA level of *BmCBP* gene in the cells was significantly down-regulated to some extent. (**B**) Compared with the protein level of α-tubulin/Histone H3, the acetylation level of H3K27 was down-regulated at 48 h after transfection of dsRNA or siRNA. (**C**) The acetylation level of the BmApoLp-II protein was significantly down-regulated by RNA interference using dsRNA or siRNA. “+” means existence, “-” means no existence. (**D**) The interaction between endogenous BmCBP protein and overexpressed His-BmApoLp-II protein in silkworms. (**E**) After co-transfection of dsRNA or siRNA, the expression of the BmApoLp-II protein in BmN cells was significantly down-regulated. The data represent the results of at least three independent experiments. * *p* < 0.05, ** *p* < 0.01, *** *p* < 0.001. The effect of RNAi using dsRNA was significantly more effective than that of RNAi using siRNA.

**Figure 5 insects-14-00309-f005:**
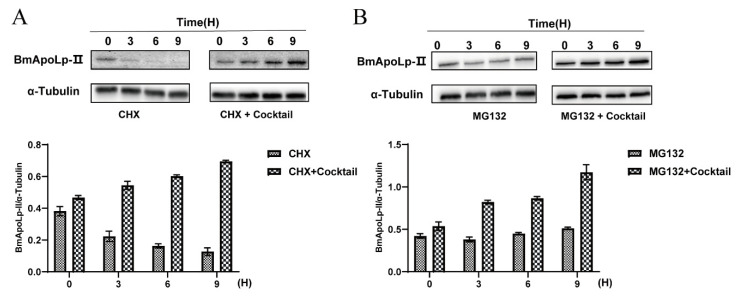
**Acetylation could improve the stability of protein in cells.** (**A**) The influence of up-regulation of acetylation by cocktail on BmApoLp-II protein degradation where CHX blocking intracellular protein synthesis pathway. (**B**) The influence of up-regulation of acetylation by cocktail on BmApoLp-II protein accumulation where MG132 is blocking the intracellular protein degradation pathway. CHX, cycloheximide; TSA, trichostatin A.

**Figure 6 insects-14-00309-f006:**
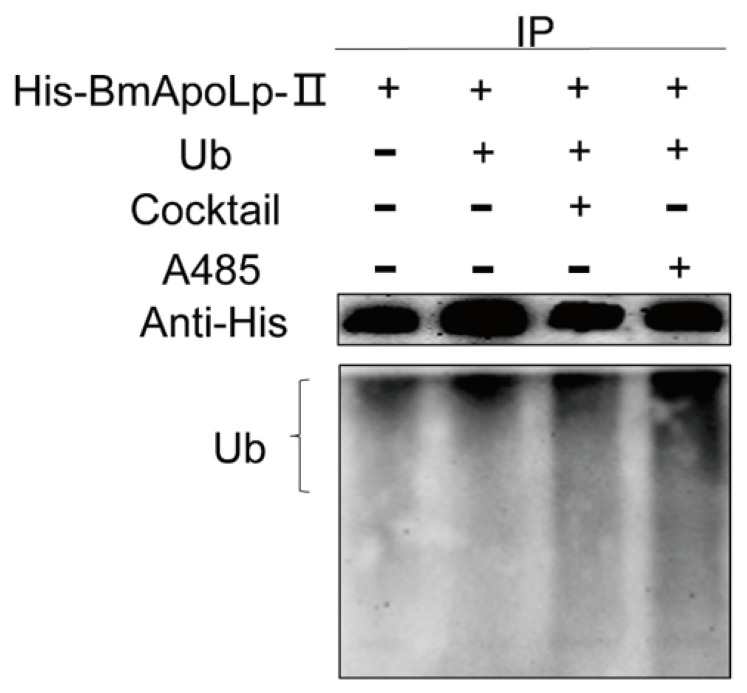
**Acetylation could compete for ubiquitination to stabilize BmApoLp-II protein.** The acetylation of the BmApoLp-II can compete for its ubiquitination modification. pIEX-1-s-GFP-BmApoLp-II and pIEX-1-si-Ub were co-transfected into BmN cells. After using cocktail to increase the acetylation level of BmApoLp-II, its ubiquitination level decreased, whereas using A485 to decrease the acetylation level of BmApoLp-II increased its ubiquitination level. “+” means existence, “-” means no existence.

## Data Availability

Data are contained within the article.

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
