# Peer review of "BmCBP Catalyzes the Acetylation of BmApoLp-II Protein and Regulates Its Stability in Silkworm, *Bombyx mori"

_insects, 2023, doi:10.3390/insects14040309_

Round 1

Reviewer 1 Report

This paper demonstrate that acetylation could improve the stability of BmApoLp-Ⅱ protein by competing its ubiquitination in silkworm, Bombyx mori. The results is meaningful. However, there are some concerns need to be addressed before publication.

1. The resolution of Figure 1A was very low.

2. In Figure 6, it seems that A485 was not used in this experiment?

Reviewer 2 Report

This is an interesting paper exploring the role of post translational modifications on the stability of selected proteins, and on the expression of selected genes. The authors should be commended for the amount of work presented here.

General Comments:

There are numerous grammatical and syntax errors throughout the paper that take away from the context of the paper and the results. The next iteration of the paper should be reviewed by a protein chemist or molecular biologist whose native language is English.

In many of the sections of the methods there are not enough details to allow someone to repeat the experiments completely. These should be expanded. In sections such as the qPCR section the authors state that the relative level of the target gene was calculated by 2-DDcT, but there is no explanation of this or a reference.

In the results sections sometimes the authors indicate what is not obvious in the figures. It is difficult to see the band of interest in Fig 2A or 2D. It is not clear in the figures how the grey scale analysis of Image J was used to show differences, nor how these differences were analyzed statistically. Similarly in Fig 3 H and I, rather than put in a figure that states there was no “meaningful” change in mRNA expression, simply put a statement in the text that there were no statistically significant differences in the mRNA levels. This will reduce the number of figures. Every time the authors state there is a statistical difference in something, the P value should be included.

In figure 4 A the authors indicate a “significant” reduction in mRNA expression as a result of RNAi or siRNA impacts. While they measured a statistically significant difference, how might this affect the role of this gene/protein on insect physiology? Not all mRNAs are translated to functional protein; so what might the biological implications be?

Also in the results section there is a repeat of many of the Methods used. If the Methods section is detailed from the beginning, and the sections have the same title in Methods and results there is only the need to insert the results, not a repeat of the methods.

A lot of science went into this work, and a lot of hours of lab time. Yet the authors have only about half a page of discussion. The first half is similar to the introduction. The second part is a summary of the data, but does not use any references nor does it relate the research findings to other studies and the state of the current understanding of these processes in the insect physiology world.        

Reviewer 3 Report

In the submitted manuscript the authors have identified a potential enzyme with acetylase activity towards Apolipoprotein II, an insect nutrient storage protein in Bomyx mori. The authors have used a series of enzyme inhibitors to support the acetylation of the Apolipoprotein and the contribution of acetylation to the stability and prevention of degradation by proteosomes. Authors have also knocked down the expression of CBP, an enzyme that could potentially acetylate ApoLp II.

Authors have done an incredible job with all the biochemical experiments performed with sufficient number of replicates. The experiments provide a comprehensive view on the acetylation of ApoLp II. The manuscript is well written given the complexity of the data and the experiments performed. 

However, the authors claim CBP is the enzyme that acetylates ApoLp II based on the RNAi and the expression of ApoLp II. The current study lacks the knowledge on the ability of CBP to perform acetylation. Therefore, an enzyme assay to show the ability of CBP to acetylate is required to support the observation made by the authors. Additionally, if an interaction of CBP and ApoLp II could be shown supports that CBP is indeed involved in acetylation of ApoLp II. 

Please find specific comments in the annotated manuscript.

Author Response

Response to Reviewer 3 Comments

In the submitted manuscript the authors have identified a potential enzyme with acetylase activity towards Apolipoprotein II, an insect nutrient storage protein in Bomyx mori. The authors have used a series of enzyme inhibitors to support the acetylation of the Apolipoprotein and the contribution of acetylation to the stability and prevention of degradation by proteosomes. Authors have also knocked down the expression of CBP, an enzyme that could potentially acetylate ApoLp II.

Authors have done an incredible job with all the biochemical experiments performed with sufficient number of replicates. The experiments provide a comprehensive view on the acetylation of ApoLp II. The manuscript is well written given the complexity of the data and the experiments performed. 

Response: Thank you for your assessment.

However, the authors claim CBP is the enzyme that acetylates ApoLp II based on the RNAi and the expression of ApoLp II. The current study lacks the knowledge on the ability of CBP to perform acetylation. Therefore, an enzyme assay to show the ability of CBP to acetylate is required to support the observation made by the authors. Additionally, if an interaction of CBP and ApoLp II could be shown supports that CBP is indeed involved in acetylation of ApoLp II.

Response: Thanks for your suggestion. CBP has intrinsic HAT activity in the past studies. And in the Bombyx mori, it was demonstrated that BmCBP catalyzed the acetylation of histone H3(H3K27)(see reference [25] and section 3.4). At the same time, we have verified it in this study. In Figure 4B, it was also confirmed by the identification of decreased acetylation level of known substrate H3K27 of BmCBP. To identify the HAT involved in BmApoLp-Ⅱ acetylation,we designed Co-IP assay. It was demonstrated BmCBP was interacted with BmApoLp-Ⅱ. Furthermore, RNAi to BmCBP (dsRNA-BmCBP/siRNA-BmCBP) down-regulated the acetylation of BmApoLp-Ⅱ, showing that BmCBP is an acetyltransferase of BmApoLp-Ⅱ protein.

 Please find specific comments in the annotated manuscript.

Response: Thanks very much for your revision and comments. We have revised them according to the comments carefully. Replies to comments are available in the PDF version.
